# Is Parent Engagement with a Child Health Home-Based Record Associated with Parents Perceived Attitude towards Health Professionals and Satisfaction with the Record? A Cross-Sectional Survey of Parents in New South Wales, Australia

**DOI:** 10.3390/ijerph17155520

**Published:** 2020-07-30

**Authors:** Muhammad Chutiyami, Shirley Wyver, Janaki Amin

**Affiliations:** 1Faculty of Human Sciences, Macquarie University, Sydney 2109, Australia; shirley.wyver@mq.edu.au; 2Faculty of Medicine and Health Sciences, Macquarie University, Sydney 2109, Australia; janaki.amin@mq.edu.au

**Keywords:** home-based records, parent engagement, child health, health professionals, primary healthcare

## Abstract

We examined parent views of health professionals and satisfaction toward use of a child health home-based record and the influence on parent engagement with the record. A cross-sectional survey of 202 parents was conducted across New South Wales (NSW), Australia. Bivariate and multivariate logistic regressions were conducted to identify predictors of parent engagement with the record book using odds ratio (OR) at 95% confidence interval (CI) and 0.05 significance level. Parents reported utilizing the record book regularly for routine health checks (63.4%), reading the record (37.2%), and writing information (40.1%). The majority of parents (91.6%) were satisfied with the record. Parents perceived nurses/midwives as most likely to use/refer to the record (59.4%) compared to pediatricians (34.1%), general practitioners (GP) (33.7%), or other professionals (7.9%). Parents were less likely to read the record book if they perceived the GP to have a lower commitment (Adjusted OR = 0.636, 95% CI 0.429–0.942). Parents who perceived nurses/midwives’ willingness to use/refer to the record were more likely to take the record book for routine checks (Adjusted OR = 0.728, 95% CI 0.536–0.989). Both parent perceived professionals’ attitude and satisfaction significantly influenced information input in the home-based record. The results indicate that improvements in parent engagement with a child health home-based record is strongly associated with health professionals’ commitment to use/refer to the record during consultations/checks.

## 1. Introduction

In 2018, the World Health Organization (WHO) endorsed universal use of home-based records in addition to facility-based records to improve maternal, infant, and child health [1]. The WHO review noted the need for further research studies on provider behavior toward use of the health records as well as best way to design the records to capture useful information that ease use by parents/professionals among others. Following the WHO review, Brown et al., [2] added that improvements are needed in health workers’ utilization of home-based record books when working with care givers/parents, including; requesting, referencing, and updating the record. Accordingly, parents and care givers of children must be capable of retaining such records and be committed to take it to the point of service to improve potential for a good start in children’s early life [3].

Growing evidence indicates that a healthy early start in children’s life has the potential to affect future mental and physical wellbeing [4,5,6] as well as educational attainment [7]. Therefore, supporting parents from pregnancy through early start in life is not only significant for empowering parents, it supports health promotion for children in the future. Family-focused interventions delivered through primary healthcare settings have the ability to enhance child public health [8]. Accordingly, families tend to consult their primary healthcare providers including family physicians, pediatricians, family/community nurses and midwives among others, for advice about their children’s development and health [8]. Improving parent–professional relationship/communication is therefore essential for building strong partnerships between parents and health professionals [9,10]. One key area to improve such parent–professionals partnership is through the use of a home-based record, which provide frontline health practitioners with a comprehensive standardized patient health history, necessary to make informed decisions [2].

Child-related home-based record have been used in various countries including Australia [11,12,13,14,15,16]. These records which serve various critical roles such as keeping up to date with immunization [2], continuity of care [17], safe pregnancy practice/delivery by skilled birth attendant and knowledge of child healthcare [18]. In our previous review [19], which informed this study, we examined usefulness of the record on child health outcomes, parent knowledge, and documentation of maternal and child health data. The review identified that a vast majority of parents (average 72%) value the child health home-based record and over 60% were satisfied with using the record, yet some parents (average 40%) reported an unsupportive attitude of health professionals toward use of the record [19]. To our knowledge, no study has specifically assessed the association of these variables (satisfaction and perceived attitude) with parent engagement with the record book. This study therefore aims to examine parent views of health professionals and satisfaction toward use of child health home-based record and the influence on parent engagement with the record book.

### Hypothesis

As noted above, the hypothesis tested in this study is explicitly based on the findings of our systematic review study [19].
Parent engagement with a child health home-based record is associated with their perception of health professional commitment and their satisfaction toward use of the record.

The significance of testing this hypothesis is that it will inform policies regarding improvement of parent engagement with a child health home-based record. As the New South Wales (NSW) child health home-based record receives strong endorsement and promotion by government to emphasis its important role in child health, exploring voices of parents will inform local policies to enhance parent experiences of utilizing child health home-based record. Similarly, considering WHO plans to update of their guidelines for home-based records by 2023 [1], the outcome of this study will provide insight toward improving utilization of home-based records by both health professionals and parents/care givers.

## 2. Method

### 2.1. Study Design, Population and Instrument

A community-based cross-sectional survey of parents was conducted in NSW Australia. Parents of children 0–5 years of age as at 2019 were recruited across NSW. The instrument for data collection was a semi-structured questionnaire generated from our systematic review [19]. It consists of a section on sociodemographic characteristics, parent personal experiences of using the record book, parent experiences with health professionals, and parent satisfaction with the record book.

This study was approved by Macquarie University Human Research Ethics Committee with a reference no. 5201933826834.

### 2.2. Study Sample Size

Sample size was estimated using GPower software (Heinrich Heine University, Westphalia, Germany) [20]. In line with the finding of a similar study that examined predictors of children’s guardians’ engagement with a home-based record [21], to estimate an odds ratio of 1.9, a minimum sample of 129 participants was required to generate a power of 80% at an alpha value of 0.05. The minimum required sample was exceeded with a total of 202 parent–child pairs recruited for this study.

### 2.3. Data Collection

Participants were approached through childcare centers and online family websites within NSW/Australia. Only parents who read and signed the participant information and consent form were included in the study. The survey instrument was then administered to parents who voluntarily agreed to participate in the study. Participants were given the option of completing an online or hard-copy version of the survey.

Responses for all engagement and perceived professionals’ attitude questions were on a four-point Likert scale (always, usually, sometimes, never). Parents were asked to rate different aspects of engagement with the record book and commitment of various health professionals based on their personal experiences of perceived willingness of health professionals to use/refer to the record during consultations and to fill-in appropriate information. Satisfaction was assessed using domains namely: ease of reading, ease of locating information, ease of understanding the words, organization of contents, and sturdiness of the child health home-based record, each using a five-point Likert scale (not satisfied, somewhat satisfied, moderately satisfied, very satisfied, excellently satisfied).

### 2.4. Definition of Variables

Engagement with a child health home-based record (dependent variable) was defined in terms of frequency parents indicated to have taken the record book to routine child clinic visits, write/record information in the record book, and read child data/other relevant information in the record book. The predictor (independent) variables include parent perceived attitude of health professionals (general practitioners (GPs), pediatricians, nurses/midwives, other professionals) and their satisfaction with the home-based record based on the five domain areas (ease of reading, ease of locating information, ease of understanding the words, organization of contents and sturdiness). Average satisfaction from the five domain areas was computed to identify overall satisfaction with the record book. For the purpose of regression analysis, parents were considered ‘regularly’ engaging with the home-based record if they indicated they ‘always or usually’ take the record book for routine checks, read or write in the book. Engagement with the book is considered ‘not regular’ if parents indicated ‘sometimes or never’ take the book for routine checks, read or write in the book. Furthermore, parent perceived attitude of professionals was considered ‘positive’ if parents indicated professionals ‘always or usually’ used the record book. It is considered ‘negative’ if parents indicated professionals’ attitude as ‘sometimes or never’ used the book. Similarly, parents were considered ‘highly satisfied’ if they indicated ‘excellently or very satisfied’, while it was considered ‘poorly satisfied’ if they indicated ‘moderately or somewhat satisfied’.

### 2.5. Data Analysis

Data collected was coded and analyzed using SPSS statistical package (IBM Corp. IBM SPSS Statistics for Macintosh, Version 25.0. Armonk, NY, USA). Descriptive statistics using proportion, mean and standard deviation were used to summarize variables for sociodemographic characteristics and parent experiences of using the record book. Spearman rho (r) correlation coefficient was used to determine the strength of association between variables of parent engagement with the record book with variables of perceived attitude of health professionals and parent satisfaction. Bivariate logistic regression was conducted to examine the influence of individual variables on parent engagement with the home-based record. Multivariate logistic regression analysis was then performed by constructing three models, each to explore interrelated predictors affecting individual parent engagement with the record book (take the record book for routine checks, read or write in the record book). Crude odds ratio (COR) and adjusted odds ratio (AOR) were respectively used as the effect-size measures in the bivariate and multivariate analyses. A *p*-value of 0.05 or less was considered significant at 95% confidence interval (CI).

## 3. Results

### 3.1. Parents Characteristics

Of the 202 participants, the majority (89.9%) were females with an average age of 35.6 years. Youngest child age at the time of data collection ranged from 3 to 65 months with an average of 26.1 months. About 70% of the parents had at least a university degree, while others education was classified as ‘non-degree’ ranging from less than year 9 to holding a post-secondary qualification. The majority of parents (77.2%) identified their first language as English (Table 1).

### 3.2. Parent Engagement, Perception of Professionals, and Satisfaction with the Home-Based Record

As can be seen in Table 2, 63.4% of the parents always or usually take the child health home-based record for routine child checks compared to 37.2% that read or 40.1% that write information in the record book. A total of 33.7% of parents perceived GPs as always or usually use/refer to the record in comparison to pediatricians (34.1%) or nurses/midwives (59.4%). Similarly, more nurses 68.3% were perceived as always or usually recording information during routine checks compared to GPs during consultations (24.3%), pediatricians at specialist visit (28.3%) or doctors at emergency contacts (11.4%). Parents were extremely or very satisfied with ease of reading (47.6%), ease of locating information (42.1%), ease of understanding words (68.9%), ease of organization of contents (44.6%) and sturdiness of the record book (44.6%).

### 3.3. Correlation between Variables

Correlations between variables of parent engagement with perceived attitude of professionals and satisfaction are reported in Table 3 and Table 4 respectively. Significant positive correlations exist between taking the record for a routine check with parent perceived attitude of GPs or nurses to refer to the book and nurses to record information in the book. Reading the record book was significantly correlated with perceived attitude of GPs and pediatricians to refer to the record book or record information at consultations. Writing information in the record book was significantly associated with all the variables of perceived attitude except for perception of other professionals to refer to the record (*p* = 0.432) or doctors to record information at emergency contacts (*p* = 0.434). Similarly, taking the record book for routine checks was significantly correlated with satisfaction in ease of reading and organization of contents. Reading and writing information in the record was positively correlated with all variables of satisfaction except for satisfaction with sturdiness of the record book (*p* = 0.213). Overall satisfaction with the home-based record was significantly correlated with writing information in the record book but not taking for routine checks (*p* = 0.062) or reading information from the book (*p* = 0.051).

### 3.4. Regression Analysis

Table 5 shows outcome of logistic regression analysis, indicating association of parent engagement with the child health home-based record at bivariate and multivariate levels. Taking the record book for routine checks was significantly influenced by parents’ perceived attitude of nurses at both bivariate (COR = 0.723, 95% CI 0.544–0.961) and multivariate (AOR = 0.728, 95% CI 0.536–0.989) levels. This indicates that parents who perceived negative attitude of nurses to use/refer to the record book were less likely to take the book for routine child checks compared to parents who perceived positive nurses’ attitude. Similarly, reading information in the record book was significantly influenced by parents’ perceived attitude of GPs with both bivariate (COA = 0.635, 95% CI 0.440–0.918) and multivariate (AOR = 0.636, 95% CI 0.429–0.942) analyses. This implies that compared to parents who perceived positive attitude of GPs to use/refer to the record book, parents who perceived negative attitude of the GPs to use/refer to the record book were less likely to read what is written in the book. Writing information in the record book was significantly influenced by parent perceived attitude of GPs (COR = 0.542, 95% CI 0.373–0.786), pediatricians (COA = 0.719, 95% CI 0.564–0.916), and nurses (COR = 0.648, 95% CI 0.471–0.893) at the bivariate analysis. In the multivariate analysis, perceived attitude of all professionals significantly influenced parent to write information in the record book. Accordingly, compared to parents who were highly satisfied with the home-based record, parents who were poorly satisfied were less likely to use the home-based record (AOR = 0.717, 95% CI 0.517–0.993) by recording child health information.

### 3.5. Recommendation for Use of Child Health Home-Based Record

Of the participants, 115 (56.9%) recommended ways of improving utilization of child health home-based record using an open-ended question; ‘please recommend any way(s) that you think the blue book (child health home-based record) or its usage could be improved’. Many of the parents (43%) opined that it should be made electronic, which could be accessed online (as part of Australia’s My Health Record) or via a mobile app. More than a third of parents (39%) recommended re-designing the record book by reducing its bulkiness, adding more pages for unhealthy children, adding more rooms for immunization, adding more than basic information, expand eye and hearing test, remove binder and use rings, add “do’s” and “don’ts” for parents. Similarly, 14% of parents recommended encouraging health professionals using phrases such as; ‘make it mandatory for GPs’, ‘professionals ask for it’, ‘must be use by professionals particularly GPs and community nurses’ and ‘if doctors show interest’.

## 4. Discussion

This study specifically investigated the views of parents with regards to health professional behaviour and personal satisfaction, and how both factors influence parent engagement with a child health home-based record. We previously proposed association between parent engagement with the child health home-based record and early child development or first-born status [22]. In a similar manner, the finding of this study indicates a link between parent perceived attitude of health professionals and parent engagement with the record book. Despite the widespread use of various forms of home-based records, the WHO universal endorsement of the records in 2018 noted the need for further research to close the knowledge gaps to achieve its ultimate goal of promoting maternal, infant and child health. With health professionals being at the forefront of achieving this goal, the findings of this study will contribute toward emphasizing the need for frontline health professionals’ commitment in the update of the WHO guidelines by 2023.

Of the three domains of parent engagement with the home-based record assessed, parents regularly took the record book for routine child health checks (63.4%) compared to reading (37.2%) or writing (40.1%) information in the book. This may not be unconnected with the fact that one of the key rationales of providing parents with a home-based record is to take it along for all health contacts [3,14,23], which enables continuity of care [17] and improved communication between parents and professionals [24]. Similarly, parents view nurses as the most likely professionals to use/refer to the record book during consultations in comparison to GPs and pediatricians, among others. This finding is in line with previous studies on home-based records [14,17,23,25,26], of which pediatricians [23] or GPs [14,26] were reported as least likely to use the record book. However, one study conducted in United Kingdom reported higher proportion of GPs who regularly use the record book compared to children nurses and pediatricians [23]. Overall, the findings of this study support the results of our previous systematic review indicating unsupportive attitude of health professionals despite parent’s commitment to use of the record as well as its positive health impact such as immunization uptake, antenatal care, and practice of breast feeding [19]

The positive association found in this study between taking the record book for routine health checks and perceived attitude of nurses to fill-in appropriate information in the book could be explained considering the guideline of the record book in NSW. In line with the recommendation of the NSW child health home-based record (known as blue book), family health nurses, GPs, or pediatricians are charged with the responsibility of conducting regular child health and development checks from birth to 4 years [27]. This therefore enable nurses, particularly community-based nurses, to commit to recording information in the record book at routine checks. A previous study of the NSW child health home-based record [16] also reported more than three-fourth of parents who regularly take the record book for schedule checks and indicated nurses as most likely to make entries. Furthermore, another study in NSW [14] identified that parents who regularly take the record book for routine checks were more likely to take it for GP consultations, despite less commitment of most GPs to record the information. Various factors could be associated with GPs/pediatricians lower commitment to use of the home-based record such as lack of time/workload [28], which is beyond the scope of this study.

Although less than half of the parents regularly write information in the record book, the writing habit was significantly associated with both variables of perceived professional’s attitude and satisfaction. While the outcome of this study cannot precisely establish the connection between these variables, it is likely to be influenced by parent perceived approval of appropriate parenting by professionals. On the other hand, reading habit as an element of parent engagement with the home-based record was found lowest in this study, of which only 37% of parents regularly read child-related data or other information in the book. This may be associated with parents’ previous experience of using the record book, of which apart from first time parents, other parents might be familiar with the content of the record book from their previous child and hence likely to affects their reading habit. Another finding of this study indicated that parent’s commitment to read the record book is significantly associated with perceived attitude of GPs to use or refer to the record during consultations. This indicates that parents who experience their GPs/family physicians recording information or referring to the record during consultations are more likely to read what is written in the book. This finding is in contrast with other studies that indicated more than three-quarters of parents who regularly read the home-based record to find important information [16] or read the developmental section before a schedule visit [29]. In a previous study in NSW [14], 77% of parents reported GPs willingness to use/refer to the record at consultations, although it was not reported if this was associated with parents reading habit. On the other hand, other professionals including physiotherapist, were viewed as least likely to use the record based on the findings of this study. This could be explained by the fact that other professionals do not have a pre-determine requirement to fill in information about the child, but rather as the child condition demands. Moreover, no association exists between parent engagement with the record book (in-terms of taking for regular checks and reading) and perceived attitude of other health professionals. This finding is in line with a previous study [14], which found other health professionals as least likely to use the home-based record compared to nurses and GPs.

Satisfaction with the record book is generally higher among parents according to the finding of this study. Satisfaction was highest in-terms of reading and ease of understanding words but lowest with ease of locating information in the record book (Table 2). Ease of reading/understanding could be associated with the fact that the NSW home-based record was made in various languages to aid understanding [27]. However, locating information could be difficult due to the volume of the booklet and hence, the need for it to be re-organize. In contrast to a similar study in United State of America [30], satisfaction with the home-based record was highest in reading and sturdiness of the record but lowest with ease of understanding words. Accordingly, parent recommendations on use of the home-based record indicated that majority of the parents were of the opinion that the record book should be made electronic. This could be associated with parent poor satisfaction toward locating information in the hardcopy book, of which a digital version would make it easier to scroll across the book. Similarly, parent desire for a digital home-based record could be associated with global advancement in information and communication technology in healthcare. Electronic health record is not only important in terms of ease of use for both professionals and service users [31], but improvement in patient care though provision of real-time data [32], completeness, and reliability of data [31] as well as research/epidemiological studies [33].

This study’s strength was in providing important evidence about parent engagement with a child health home-based record in NSW since its major update in 2013. Similarly, the study has a strength of testing its hypothesis based on findings of a published systematic review as well as its specific focus on parent views of health professionals. Despite the strengths, caution should be taking in interpreting the findings due its limitations. The cross-sectional nature of the study makes it impossible to establish a causality between parent engagement with the home-based record and perceived professionals’ attitude or satisfaction, but rather a possible association between the variables. Similarly, parent engagement may be less than reported due to a possibility of social bias in reporting. Furthermore, the participants were recruited using a non-probability sampling technique, most of which were around Sydney and not randomly distributed across NSW. Additionally, the majority of parents recruited through childcare centers were centers associated with universities, which could explain the high percentage of parents having a university degree. Therefore, findings may not be generalizable to Australian population.

## 5. Conclusions

Parent engagement with a child health home-based record was positive, particularly with respect to taking the record book for routine child checks. Perceived attitude of health professionals (positive or negative) was found to significantly influence parent engagement with a home-based record. GPs and nurses being frontline health professionals that look after children can influence parents to read what is recorded and take the record book to the point of service, respectively. Commitment of all health professionals and satisfaction toward use of the home-based record can significantly affect parent’s willingness to use the record book by writing appropriate child health information. The overall outcome of this study answered our hypothesis that parent-perceived attitude of health professionals and satisfaction is associated with parent engagement with the record book. Therefore, to improve parent engagement with a child health home-based record and achieve its goal of child health promotion, health professionals (particularly GPs) must be committed to use or refer to the record book during child consultations/ regular checks. Further studies could investigate health professionals-perceived barriers and motivation toward use of the home-based record.

## Figures and Tables

**Table 1 ijerph-17-05520-t001:** Demographic characteristics of respondents.

Variables	Response	Total
Parent Age	35.6 ± 5.66	198
Index child age (months)	26.1 ± 17.33	194
Parent Gender		199
Females	179(89.9)	
Male	20(10.1%)	
Educational level		200
University degree or higher	139(69.5%)	
Non-degree	61(30.5%)	
First language		202
English	156(77.2%)	
Others	46(22.8%)	

**Table 2 ijerph-17-05520-t002:** Number (percentage) of reported utilization, perception and satisfaction (*N* = 202).

Variables	Frequency (%)					
	Always	Usually	Sometime	Never	N/A	
Parent engagement with record						
Take for routine check	84(41.6)	44(21.8)	35(17.3)	36(17.8)	3(1.5)	
Read	29(14.4)	46(22.8)	78(38.6)	45(22.3)	4(2.0)	
Write Information	41(20.3)	40(19.8)	70(34.7)	48(23.8)	3(1.5)	
Parent perceived attitude						
GPs to use/refer	27(13.4)	41(20.3)	79(39.1)	40(19.8)	15(7.5)	
Pediatrician to use/refer	31(15.3)	38(18.8)	43(21.3)	24(11.9)	66(32.7)	
Nurses/midwives to use/refer	69(34.2)	51(25.2)	32(15.8)	16(7.9)	34(16.8)	
Other professionals to use/refer	5(2.5)	11(5.4)	26(12.9)	62(30.7)	98(48.5)	
Nurses to record information at routine checks	107(53)	31(15.3)	28(13.9)	10(5)	26(12.9)	
GPs to record information at consultations	19(9.4)	30(14.9)	90(44.6)	49(24.3)	14(7)	
Pediatricians to record information at specialist visit	27(13.4)	30(14.9)	49(24.3)	33(16.3)	63(31.2)	
Doctors to record information at emergency	9(4.5)	14(6.9)	21(10.4)	77(38.1)	81(40.1)	
Satisfaction	**Extreme**	**Very**	**Moderate**	**Somewhat**	**No**	**N/A**
Ease of reading	30(14.9)	66(32.7)	63(31.2)	24(11.9)	6(3)	13(6.4)
Ease of locating information	25(12.4)	60(29.7)	57(28.2)	34(16.8)	12(5.9)	14(6.9)
Ease of understanding words	49(24.3)	90(44.6)	38(18.8)	10(5)	2(1)	13(6.4)
Organization of contents	29(14.4)	61(30.2)	63(31.2)	27(13.4)	7(3.5)	15(7.4)
Sturdiness of the book	30(14.9)	60(29.7)	48(23.8)	35(17.3)	16(7.9)	13(6.4)
Overall satisfaction	26(12.9)	65(32.2)	72(35.6)	22(10.9)	4(2)	13(6.4)

N/A—not available/not applicable, GPs—general practitioners.

**Table 3 ijerph-17-05520-t003:** Correlation between parent engagement with record book and perceived attitude.

Perceived Attitude	Taking Record Book for Routine Checks	Reading Record Book	Writing Information in Record Book
GPs to use/refer	0.169 *	0.229 ***	0.273 ***
Pediatrician to use/refer	0.086	0.150 *	0.261 ***
Nurses/midwives to use/refer	0.289 ***	0.116	0.221 **
Other professionals to use/refer	−0.034	0.052	−0.058
Nurses to record information at routine checks	0.320 ***	0.109	0.241 ***
GPs to record information at consultations	0.109	0.262 ***	0.230 ***
Pediatricians to record information at specialist visit	0.139	0.221 **	0.321 ***
Doctors to record information at emergency	0.032	0.066	0.057

* *p* ≤ 0.05, ** *p* ≤ 0.01, *** *p* ≤ 0.001.

**Table 4 ijerph-17-05520-t004:** Correlation between parent engagement with record book and satisfaction.

Satisfaction	Taking Record Book for Routine Checks	Reading Record Book	Writing Information in Record Book
Ease of reading	0.173 *	0.217 **	0.198 **
Ease of locating information	0.121	0.159 *	0.154 *
Ease of understanding words	0.092	0.164 *	0.168 *
Organization of contents	0.145 *	0.184 *	0.157 *
Sturdiness of the book	0.068	0.092	0.056
Overall satisfaction	0.137	0.143	0.168 *

* *p* ≤ 0.05, ** *p* ≤ 0.01.

**Table 5 ijerph-17-05520-t005:** Logistic regression examining predictors of parent engagement with child health home-based record.

Variables	Categories	Taking Record Book for Routine Checks		Reading Record Book		Writing Information in Record Book	
		Crude OR (95% CI)	Adjusted OR (95% CI)	Crude OR (95% CI)	Adjusted OR (95% CI)	Crude OR (95% CI)	Adjusted OR (95% CI)
Perceived attitude of GPs	Negative	0.749 (0.530–1.059)	0.760 (0.526–1.097)	0.635 (0.440–0.918) *	0.636 (0.429–0.942) *	0.542 (0.373–0.786) ***	0.597 (0.399–0.896) **
	Positive	Reference	Reference	Reference	Reference	Reference	Reference
Perceived attitude of Pediatricians	Negative	1.034 (0.814–1.314)	1.023 (0.776–1.348)	0.790 (0.619–1.009)	0.796 (0.606–1.045)	0.719 (0.564–0.916) **	0.637 (0.478–0.850) **
	Positive	Reference	Reference	Reference	Reference	Reference	Reference
Perceived attitude of Nurses/midwives	Negative	0.723 (0.544–0.961) *	0.728 (0.536–0.989) *	0.946 (0.708–1.263)	1.053 (0.772–1.437)	0.648 (0.471–0.893) **	0.681 (0.485–0.956) *
	Positive	Reference	Reference	Reference	Reference	Reference	Reference
Perceived attitude of other professionals	Negative	1.136 (0.835–1.546)	1.189 (0.842–1.681)	0.762 (0.560–1.038)	0.827 (0.591–1.157)	1.266 (0.924–1.735)	1.622 (1.117–2.356) **
	Positive	Reference	Reference	Reference	Reference	Reference	Reference
Overall satisfaction	Poorly satisfied	0.797 (0.594–1.071)	0.778 (0.569–1.065)	0.869 (0.650–1.161)	0.858 (0.629–1.170)	0.821 (0.616–1.094)	0.717 (0.517–0.993) *
	Highly satisfied	Reference	Reference	Reference	Reference	Reference	Reference

OR odds ratio, CI confidence interval, * *p* ≤ 0.05, ** *p* ≤ 0.01, *** *p* ≤ 0.001.

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
