# Peer review of "Is Parent Engagement with a Child Health Home-Based Record Associated with Parents Perceived Attitude towards Health Professionals and Satisfaction with the Record? A Cross-Sectional Survey of Parents in New South Wales, Australia"

_ijerph, 2020, doi:10.3390/ijerph17155520_

Round 1
Reviewer 1 Report
A well conducted and written research piece.
Line 71 - Typing error 'update of their...'
Line 119 - Ethics approval probably best sits in 2.1
Table 2 - Likert scale is different to described in 2.4 Definition of variables.
Table 2 - Satisfaction of parents - Are these generalisable to the Australian parenting population when most parents had a university education - Extreme to Very (48% = less than half)? Briefly address this limitation in discussion. Line 160 - Correlation numbers in text are not the same as in Table 3? In regard to the findings - the many interesting correlations with writing in the book revealed in Tables 3 and 4 should be interpreted in the discussion. Perhaps parents are more likely to write in the book when they have reason to believe it will be read for social approval of appropriate parenting from health professionals. Perhaps writing in the book makes them more invested in the process (internal locus of control) and then more likely to actively place the book in health professionals possession for use/referral... perhaps there may be another underlying connection worth exploring in the future. The encouragement for parents to write in the book (parent engagement) appears it may be as important as the interactions with and influence of health professionals in future efforts of child health promotion.
Line 196 - Try and avoid starting a paragraph with a number. 'A total of 116...'
Line 219-221 - Please re-state percentages for the ease of the reader.
264 - As stated earlier Parent Satisfaction reflects experiences from 69% of parents with a university level education.
289 - Parent engagement may actually be less as social bias may be present in their reporting.
Please attend to table alignment throughout.
Thank you for the read. Very interesting research that provides valuable information in child health promotion efforts towards maximising the benefits of the child health home-based record books.
Reviewer 2 Report
Thank you for your submission. Excellent work overall. It appears that your conclusions are well backed up by the data you collected, and that you fairly analyzed the data and correctly interpreted the meaning of the data. As a general practitioner myself, what I would be most interested in seeing you discuss more extensively is any research showing specifically that the record book improves health. Logically it makes sense, however, is this backed up by hard data?
In the introduction, you emphasize how engagement tends to improve outcomes, but I did not specifically see where you discuss that engagement with a home record book improves outcomes. Is your point simply that the record book helps improve engagement with the healthcare professional, and that this improved engagement leads to better outcomes? If yes, then anything that improves engagement would lead to better outcomes, and the record book itself isn't necessary.
Alternatively, is there something special about a record book? Is engagement with a healthcare professional specifically utilizing a record book important?
Until you can show that engagement with the record book is important, your research has very little clinical importance, other than the WHO recommends the book. I want data, not just an organization's endorsement. So anything you can reference besides simply a WHO recommendation would be helpful. Why does the WHO recommend engagement with a record book? What is their recommendation based upon? What does the data show?
If you can convince the reader that 1) a record book is important, and 2) engaging with a healthcare provider over the record book is important, then it logically follows that 3) your research is critically important.
In the paragraph starting on line 51, you clearly make a case for the record book. You reference your previous review, but only state that parents valued the record and were satisfied, but what evidence points to the home based record improving clinical outcomes? Does the home based record only create satisfied parents? Or does it also improve the health of the child?
I believe that it is highly likely that the home based record improves the health of children, but I do not have any hard data to support my opinion. I was hoping that your manuscript would reference some hard scientific data supporting its use.
Overall, excellent work. Please include a brief discussion of the research showing that a record book increases clinical outcomes. Parent satisfaction isn't enough because parents could be satisfied with all sorts of alternatives as well. Why is the home based record book so important? Can you prove it?
Thank you again for your great research. This is an important area of research and I hope you continue your work in this area.
